# Diagnostic system for the detection of severe fever with thrombocytopenia syndrome virus RNA from suspected infected animals

Eun-sil Park[1], Osamu Fujita[1], Masanobu Kimura[1], Akitoyo Hotta[1], Koichi Imaoka[1], Masayuki Shimojima[2], Masayuki Saijo[2], Ken Maeda[1,3], Shigeru Morikawa[1,4]*

1 Department of Veterinary Science, National Institute of Infectious Diseases, Shinjuku, Tokyo, Japan, 2 Department of Virology I, National Institute of Infectious Diseases, Shinjuku, Tokyo, Japan, 3 Laboratory of Veterinary Microbiology, Yamaguchi University, Yamaguchi, Japan, 4 Department of Microbiology, Faculty of Veterinary Medicine, Okayama University of Science, Imabari, Ehime, Japan

* morikawa@nih.go.jp

**Data Availability Statement:** All sequence files are available from the GenBank database (accession

## Abstract

### Background

Severe fever with thrombocytopenia syndrome virus (SFTSV) causes severe hemorrhagic fever in humans and cats. Clinical symptoms of SFTS-infected cats resemble those of SFTS patients, whereas SFTS-contracted cats have high levels of viral RNA loads in the serum and body fluids. Due to the risk of direct infection from SFTS-infected cats to human, it is important to diagnose SFTS-suspected animals. In this study, a reverse transcription polymerase chain reaction (RT-PCR) was newly developed to diagnose SFTS-suspected animals without non-specific reactions.

### Methodology/principle findings

Four primer sets were newly designed from consensus sequences constructed from 108 strains of SFTSV. A RT-PCR with these four primer sets successfully and specifically detected four clades of SFTSV. Their limits of detection are 1–10 copies/reaction. Using this RT-PCR, 5 cat cases among 56 SFTS-suspected animal cases were diagnosed as SFTS. From these cats, IgM or IgG against SFTSV were detected by enzyme-linked immunosorbent assay (ELISA), but not neutralizing antibodies by plaque reduction neutralization titer (PRNT) test. This phenomenon is similar to those of fatal SFTS patients.

### Conclusion/significance

This newly developed RT-PCR could detect SFTSV RNA of several clades and from SFTS-suspected animals. In addition to ELISA and PRNT test, the useful laboratory diagnosis systems of SFTS-suspected animals has been made in this study.

numbers LC514461, LC514462, LC514463, LC514464, LC514465).

**Funding:** The work was partly supported by Japan Agency for Medical Rersearch and Development (AMED) under Grant Number 20fk0108069, 20fk0108081 and Grants-in-Aid for Scientific Research under grant Number 19K06395. The funders had no role in study design, data collection and analysis, decision to publish, or preparation of the manuscript.

**Competing interests:** The authors have declared that no competing interests exist.

## Introduction

Severe fever with thrombocytopenia syndrome (SFTS) is an emerging viral hemorrhagic fever that was first identified in China [1], with cases since reported in Japan, South Korea, Vietnam and Taiwan [2–5]. Seven to eight clades of SFTSV are reportedly spreading throughout Japan, China and South Korea [6]. SFTS virus (SFTSV) has been identified as a causative virus of SFTS belonging to the genus *Banyangvirus*, family *Phenuiviridae*, order *Bunyavirales*.

SFTS has been considered to be mainly transmitted by tick bites. Ticks infest a variety of animals, while viral RNA and antibodies against SFTSV have been detected in wild animals, domestic animals and companion animals, such as dogs and cats [7–11]. Since these animals show no clinical symptoms, they have been considered subclinically infected with SFTSV. In recent years, SFTS patients without a history of tick bites have been reported, and they are considered to have received the virus through transmission from animals, such as cats and dogs [10, 12]. Furthermore, it has been shown that cheetahs [13], cats [14] and dogs can contract SFTS (*manuscript in preparation*). Thus, it is important to diagnose SFTS-suspected animals.

In Japan, conventional one-step RT-PCR, ELISA and the isolation of SFTSV are performed to diagnose human SFTS cases [15]. However, there are some limits with this system due to the non-specific reactions in case of animal SFTS cases. Thus, in this study, a reverse transcription polymerase chain reaction (RT-PCR) was developed to establish a laboratory diagnosis system for detecting several clades of SFTSV in the specimens of SFTS-suspected animals.

## Materials and methods

### Serum samples

The samples, including serum and oral and rectal swabs, were collected from SFTS-suspected cats (n = 36) and dogs (n = 19) at veterinary hospitals throughout Japan for routine SFTS diagnostic work from August 2017 to March 2019. Cats exhibiting a fever (>39 ˚C), leukocytopenia (<2000 /μL), thrombocytopenia (<200,000 /μL) and elevated levels of AST, ALT and CK were suspected of having SFTS. Dogs showed anorexia, depression, fever and some gastrointestinal tract symptoms, including diarrhea and vomiting. Clinical information was provided by veterinarians.

### A phylogenic analysis for primer design

The nucleotide sequences of 100 strains of SFTSV S segment and M segment were selected randomly from Genbank to cover all the clades and aligned and phylogenetically analyzed (S1 Fig). In brief, phylogenic trees were constructed using with the maximum likelihood method with the Tamura-Nei model using the MEGA 7 software program [16]. The robustness of the resulting branching patterns was tested using the bootstrap method with 1,000 replicates. From this analysis, it was confirmed that the seven to eight clades of SFTSV correlate with their geographical location, as has been reported previously [6]. The nucleotide identity, determined using the Bioedit sequence alignment editor [17], was 94.1%-99.1% in S segment, and 93%-99.7% in M segment among clusters. The consensus sequences of S segment and M segment among the different strains were selected using the Bioedit program, and primers were designed by the NCBI Primer-BLAST [18] from the consensus sequences.

### RT-PCR

RNAs from three strains of SFTSV belonging to different clades—SPL010 (J1 clade, accession No. AB817999), cat#1 (C4 clade, accession No. DRA007207) and HB29 (C3 clade, accession

No. NC_018137)—were used as positive controls. The copy numbers of RNA samples used as positive controls have been measured by real time RT-PCR according to our previous study [14]. Then, RNAs were diluted by 10-fold with diethylpyrocarbonate (DEPC)-treated distilled water (DW) (Nippon Gene Co., Ltd., Tokyo, Japan). DEPC-treated DW was used as a negative control. RNA was extracted using a High Pure Viral RNA Kit (Roche, Mannheim, Germany) as previously reported [14]. RT-PCR was performed using the Superscript III one-step RT-PCR system with platinum *Taq* DNA polymerase (Invitrogen, Carlsbad, CA, USA) under the following conditions: RT at 55 ˚C for 30 min, inactivation at 95 ˚C for 2 min, and then 40 cycles of PCR at 94 ˚C for 30 s, 52 ˚C for 30 s, and 68 ˚C for 30 s, followed by extension at 68 ˚C for 5 min.

## SFTSV detection from samples using four designated primers

Total RNAs were extracted from the specimens of dogs and cats using ISOGEN (Wako, Osaka, Japan) and a precipitation carrier (Ethachinmate; Wako), according to the manufacturer's instructions. RT-PCR was performed using a Superscript III one-step RT-PCR system with platinum Taq DNA polymerase (Invitrogen) with four sets of specific primers. Samples with more than two positive bands were considered SFTSV RNA-positive.

## Amplification of viral genome of S segment and phylogenetic analysis

Sequencing and phylogenetic analysis was performed on the viral S gene segment for all RT-PCR positive samples. RT-PCR was performed using a Superscript III one-step RT-PCR system with platinum Taq DNA polymerase (Invitrogen) with primers covering the entire S segment region according to a previously reported study [6]. The PCR products were determined by electrophoresis on 1% agarose gels with GR Red Loading Buffer (GRR-1000, Bio-Craft, Tokyo, Japan). The PCR products were then extracted and purified using an illustra™ GFX™ PCR DNA and Gel Band Purification kit (GE Healthcare, Buckinghamshire, UK). The samples were sequenced using the general Sanger sequencing technique.

The nucleotide sequences determined in this study were deposited in the DDBJ GenBank databases. For the phylogenetic analysis, three nucleotide data points per cluster were selected. The sequence alignment was computed using the Clustal W program of MEGA 7 software program. The phylogenetic tree was constructed using the maximum likelihood method based on the Tamura-Nei model of the MEGA program. The confidence of the tree was tested using 1000 bootstrap replications.

## Detection of IgM and IgG in cats by an enzyme-linked immunosorbent assay (ELISA)

Antibodies against SFTSV were detected by an ELISA, essentially performed as in the previously described study [19]. In brief, SFTSV- or mock-infected Huh7 cells were lysed in 1% NP40 in phosphate-buffered saline (PBS), ultraviolet (UV)-irradiated to completely inactivate SFTSV, and then clarified by centrifugation at 12,000 rpm for 10 min. The lysates were coated onto the ELISA plate (Nunc-Immuno™ plate; Thermo Fisher Scientific, Roskilde, Denmark). The antigen-coated wells were then blocked with 20% Blocking One (Nacalai Tesque, Inc., Kyoto, Japan) in PBS (blocking solution) at room temperature for 1 h. Sera of cats and dogs were inactivated at 56 ˚C for 30 min and serially 4-fold diluted from 1:100 to 1:6400 in the blocking solution at 37 ˚C for 1 h. Horseradish (HRD)-conjugated goat anti-feline IgG Fc and HRD-conjugated goat anti-feline IgM (Novus biologicals), and HRD-conjugated goat anti-dog IgM(μ) and HRD-conjugated sheep anti-dog IgG(H) were used to detect IgM and IgG antibodies in cats and dogs, respectively. The reaction was finally

visualized by a substrate for HRP (ABST, 2, 2azinobis (3-ethylbenzthiazolinesulfonic acid); Roche, Mannheim, Germany) for 30 min at room temperature. The optical density (OD) at 405 nm was measured with an iMark™ microplate reader (Bio-Rad, Tokyo, Japan). The OD values in the mock-antigen coated well were subtracted from the OD value in the respective SFTSV-antigen coated wells. The cut-off OD value was set as the average subtracted OD value plus three times the standard deviation (SD), that is, mean + 3SD, of SFTS-negative serum that had been confirmed by an indirect immunofluorescent antibody assay using SFTSV-infected Vero cells. The sera were considered positive when the OD values were above the cut-off value.

### The 50% plaque reduction neutralization titer (PRNT$_{50}$)

The PRNT test was performed to determine the neutralizing antibodies against SFTSV using Vero cells (ATCC), according to previously reported studies. Approximately 100 plaque-forming units of the HB29 strain of SFTSV were mixed with serially diluted heat-inactivated sera and incubated for 1 h at 37 ˚C and then inoculated into confluent monolayers of Vero cell in 12-well plates for 1 h at 37 ˚C. The inocula were removed, and the cells were washed once with DMEM containing 2% FBS and kanamycin and then cultured at 37 ˚C in a 5% $CO_2$ incubator with DMEM containing 2% FBS, Kanamycin and 1% methylcellulose for 1 week. Cultured cells were fixed with 10% buffered formalin and exposed to UV radiation to inactivate the virus. The cells were permeabilized with 0.1% Triton X-100, followed by incubation with rabbit antibodies against SFTSV-N as primary antibodies and HRP-conjugated recombinant protein A/G (Cat. No. 32490, Thermo Scientific, Rockford, IL, USA) as secondary antibodies. Plaques were visualized with 3, 3'-diaminobenzidine tetrahydrochloride (Peroxidase stain DAB kit [Brown stain]; Nacalai Tesque). The PRNT50 value was determined as the reciprocal of the highest dilution at which the number of the plaques was below 50% of the number calculated without cat serum.

## Results

Four primer pairs (2 for the S segment and 2 for the M segment) successfully detected the RNAs of the three SFTSV strains belonging to different clusters with a detection limit of 1–10 copies/reaction (Fig 1 and Table 1). For the detection of SFTSV RNA from samples, these four primer sets were used.

From August 2017 to March 2019, 56 cases were collected, and RT-PCR was performed to detect SFTSV RNA (Fig 2). Among them, SFTSV RNA was detected in the sera of five cats. The PCR products were confirmed with RT-PCR using all four sets of primer pairs (Fig 3). Five SFTS contracted cats showed clinical symptoms including depression, loss of appetite and jaundice. They showed body weight loss, fever, leukocytopenia (2180~7540/μL), thrombocytopenia (15,000~120,000/μL) and high total bilirubin level. All of the five cats were kept both indoor and outdoor and four of the five cats had tick-bite history. Samples were collected within one week after disease onset. Two cats were dead and three cats recovered (Table 3). The positive samples were evaluated to determine the nucleotide sequence of the S segment (Table 2). These nucleotide sequences of the S segment from five cases were phylogenetically analyzed with the corresponding segment of the Heartland virus as an outgroup (Fig 4). As a result, four strains were clustered into genotype J1, and one strain was clustered into genotype J3 of the Japanese clade. Totally, RT-PCR with four primer pairs successfully and specifically detected SFTSV belonging to four clades, including positive controls.

In addition to SFTSV RNA, antibodies against SFTSV were detected in the sera of the five cats that were PCR-positive (Table 3). Three samples had IgM and IgG against SFTSV, and

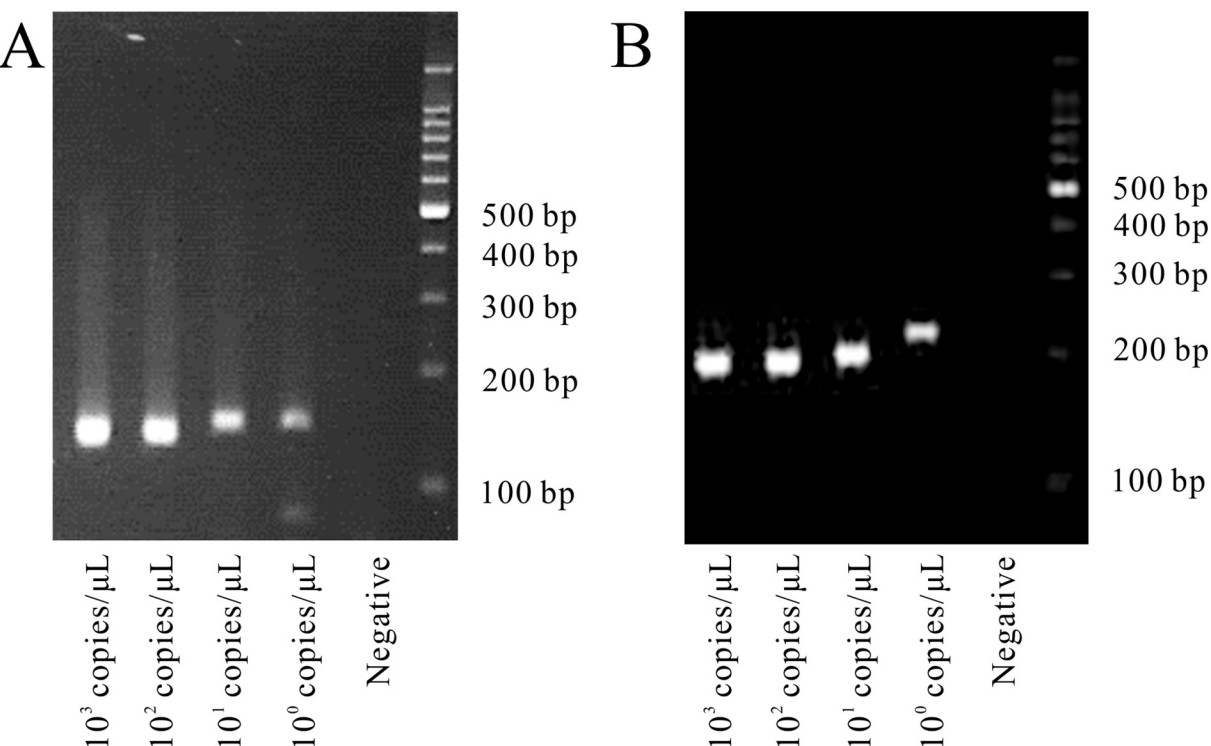

**Fig 1. RT-PCR electrophoresis for the confirmation of the detection limit.** (A) Primer 2. Viral RNA was extracted from the SPL010 strain. $10^3$–$10^0$ copies/reaction and mock sample (from left to right). A PCR product of 125 bp was observed. (B) Primer 4. Viral RNA was extracted from HB29 strain. $10^3$–$10^0$ copies/reaction and mock sample (from left to right). A PCR product of 179 bp was observed.

**Table 1. Information of primers.**

| No. of primer | Segment | Site | Product size (bp) | Nucleotide sequence | Limit of detection |
|---|---|---|---|---|---|
| 1 | S segment | 1347–1369 | 201 | 5'-TGCTGCAGCACATGTCCAAGTGG-3' | 1~10 |
| | | 1524–1496 | | 5'-GACACAAAGTTCATCATTGTCTTTGCCCT-3' | |
| 2 | S segment | 1028–1048 | 125 | 5'-GCCATCTGTCTTCTTTTTGCG-3' | 1~10 |
| | | 1131–1152 | | 5'-AGTCACTTGCAAGGCTAAGAGG-3' | |
| 3 | M segment | 2422–2442 | 185 | 5'-AGGCAAGGTTGGAGAGATACA-3' | 1~10 |
| | | 2586–2606 | | 5'-CCCCAATAGTGGTGGGTATGG-3' | |
| 4 | M segment | 373–393 | 179 | 5'-AGTTCCTGGGCCTTCATACAA-3' | 1~10 |
| | | 530–551 | | 5'-CATCACCTATCCAGAGAACCCT-3' | |

two had IgM or IgG, respectively (Table 3). Serum samples were collected at a one-week interval from one case. IgM was detected in these two interval sera, and IgG was detected in the serum collected one week later. This seroconversion pattern was similar to that of our previous study. Antibodies were not detected in the RT-PCR-negative animals.

The neutralizing antibodies against SFTSV were then measured with the $PRNT_{50}$ according to our previous study [14]. The titer of neutralizing antibodies was below the limit of detection, indicating that the antibodies detected by the ELISA could not neutralize SFTSV, similar to those of fatal human cases.

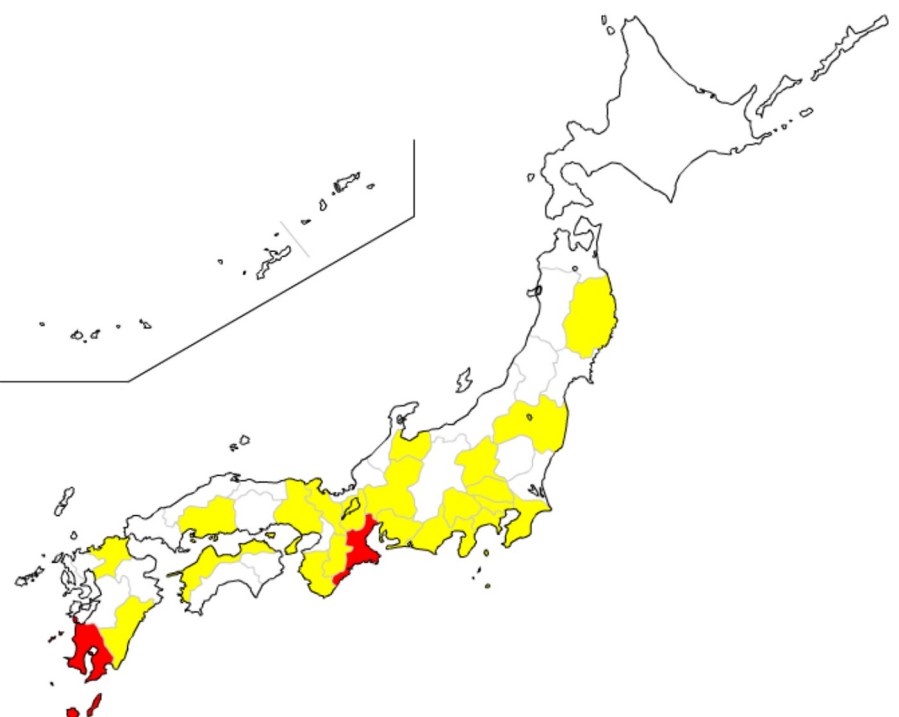

**Fig 2. Location of sample collected and diagnosed as SFTS.** The map indicates the provinces/prefectures (yellow) where samples were collected and where cats were diagnosed as SFTS (red).

## Discussion

In Japan, for the diagnosis of human and animal SFTS cases, conventional one-step RT-PCR to detect the SFTSV genome, ELISA to detect the antibodies against SFTSV and the isolation of SFTSV are performed [15]. However, non-specific reactions were obtained from some SFTS-suspected animal cases, especially dogs, in the one-step RT-PCR using primers for human SFTS diagnosis. For the simple, rapid and specific detection of SFTSV RNA from SFTS-suspected animal cases, primer sets were newly designed in this study. Four primer sets were able to detect SFTSV RNA belonging to four genotypes with a low detection limit. Some genotypes were not tested since these were not available. However, the primers used in this study might detect other clades since the sequence of these was identical to the primer sequences. Two pairs were specific for the S segment, and two pairs were specific for the M segment. SFTSV RNAs were detected from five cases using these primers. Four positive bands were observed in all five cases. These positive cases were distributed in the same region where human SFTS cases have been reported (Table 4). The sites at which two cases were detected were close to each other. The genotype of these strains was J1 and J3, showing 99.3%-99.7% homology. These findings are believed to establish the hot spot and circulation of SFTSV among ticks and animals. In Japan, several phylogenetic clades of SFTSV are circulating in the SFTS endemic region, thus it is of interest to clarify if the genomic reassortment event occurred in animal derived SFTS viruses in future. The ages of the cats ranged from 9 months to 15 years old. The period of disease onset was from January to October. All of these cats were kept both indoors and outdoors. In addition, four cases had a tick-bite history, indicating the transmission of SFTSV by tick.

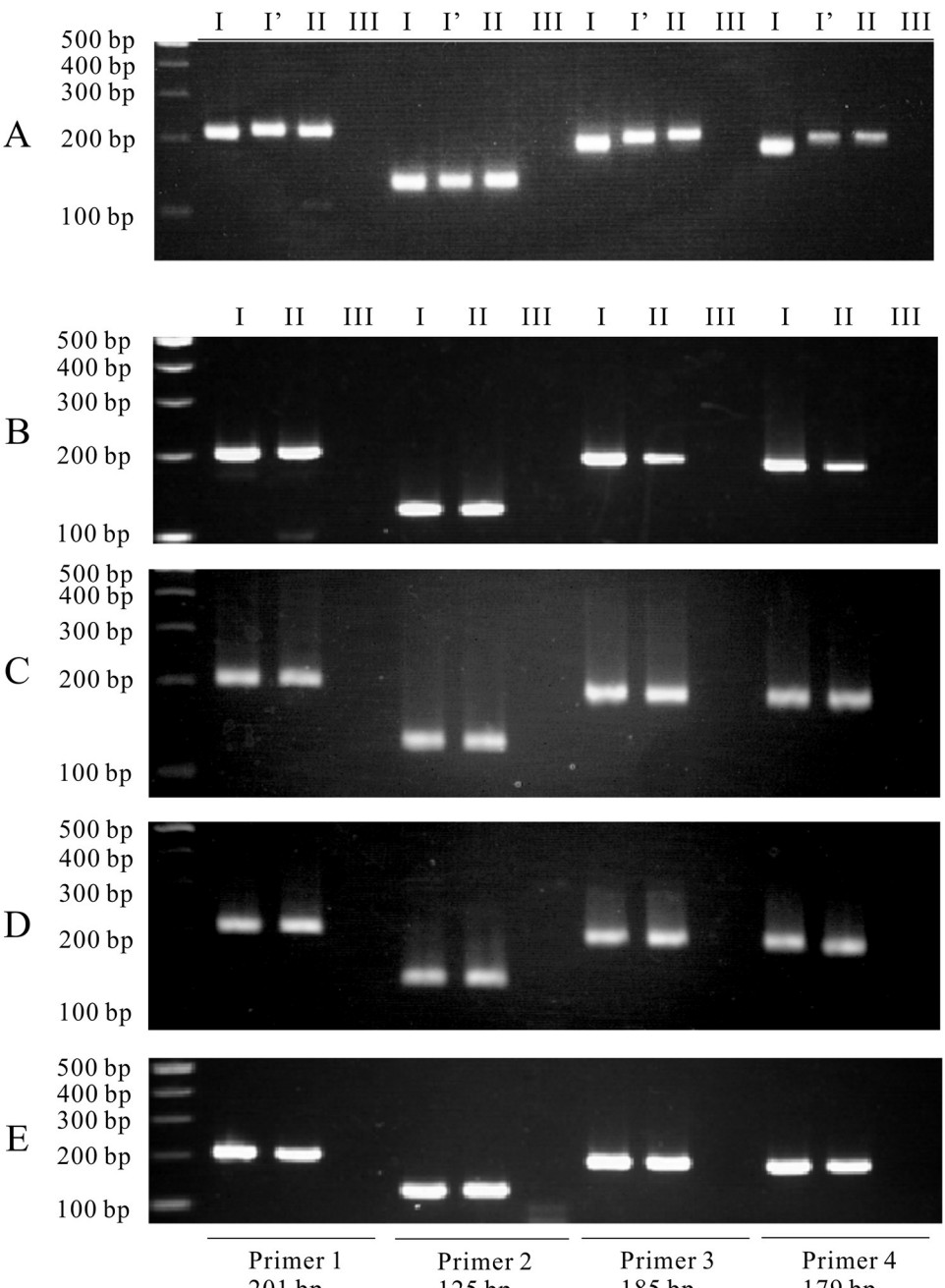

**Fig 3. RT-PCR electrophoresis of five SFTS cat cases.** The PCR product bands were observed. I and I': sample RNA, II: positive control, III: negative control. (A) No. 1, (B) No. 2, (C) No. 3, (D) No. 4, and (E) No. 5.

Their clinical symptoms were similar to those described in previous reports. Thus, these cats were diagnosed with SFTS. After cats were diagnosed as SFTS, veterinarians were informed; 1) they should wear personal protective equipment (PPE), such as gloves, mask, face shield, goggle and gown, to treat SFTS-diagnosed cats since SFTSV could be secreted from blood, saliva, urine and feces for about two or three weeks, 2) they should consult a

**Table 2. The accession number of positive cases.**

| Strain | The accession No. |
|--------|-------------------|
| JDVS17 | LC514461 |
| JDVS22 | LC514462 |
| JDVS26 | LC514463 |
| JDVS41 | LC514464 |
| JDVS47 | LC514465 |

doctor if they have a direct contact with these blood, saliva, urine and feces and then have clinical symptoms, such as, fever, 3) they should inform owners of the animals not to contact without PPE.

Five cats had IgM and/or IgG against SFTSV, determined by an ELISA. However, the titer of neutralizing antibodies was below the limit of detection. The serum specimens of the cats were collected within one week after the onset. It takes approximately two weeks to see a

**Table 3. Information of five cats.**

| No. of case | 1 | 2 | 3 | 4 | 5 |
|-------------|---|---|---|---|---|
| SFTSV RNA | + | + | + | + | + |
| IgM | + (1:400) | + (1:400) | - | + (1:1600) | + (1:100) |
| IgG | + (1:6400) | + (1:100) | + (1:100) | + (1:1600) | + (1:100) |
| Province | Kagoshima | Mie | Mie | Mie | Mie |
| Animal | cat | cat | cat | cat | cat |
| Disease onset | January 2018 | May 2018 | June 2018 | August 2018 | October 2018 |
| Sampling time point | Unknown | after 4 days | after 7 days | after 4 days | after 6 days |
| Age | 9 m | 10 m | 3 y | 14 y | 15 y 5 m |
| Sex | ♂ | ♂ | ♀x | ♀ | ♂ |
| Condition | indoor & outdoor | indoor & outdoor | indoor & outdoor | indoor & outdoor | indoor & outdoor |
| Body weight (kg) | N.D. | 2.9 | 2.8 | 2.8 | 4.9 |
| Body temperature (˚C) | 39.3 | 39.1 | >40 | 38.3 | 39.4 |
| RBC ($10^4$/μl) | 301 | 101.5 | 1191 | 831 | 388 |
| WBC (/μl) | 4000 | 2180 | 4200 | 7540 | 3600 |
| Platelet (/μl) | 56000 | 15000 | 44000 | 120000 | 18000 |
| ALT/GPT (I/U) | 80 | 69 | 59 | N.D. | 135 |
| AST/GOT (I/U) | N.D. | N.D. | N.D. | N.D. | N.D. |
| CK (I/U) | N.D. | N.D. | N.D. | N.D. | N.D. |
| T-bil (mg/dl) | 5.9 | N.D. | 2.9 | N.D. | 2 |
| FeLV | negative | N.D. | N.D. | N.D. | negative |
| FIV | negative | N.D. | N.D. | N.D. | positive |
| Symptoms | Depression, Anorexia, Intussusception (ileum to colon), GI hemorrhage, vomit, jaundice | Depression, Anorexia, jaundice | Depression, Anorexia, jaundice | Depression, Anorexia, jaundice | Depression, Anorexia |
| Tick-bite history | tick-bite scar | negative | positive | positive | positive |
| Prognosis | Dead | Recovered | Recovered | Recovered | Dead |

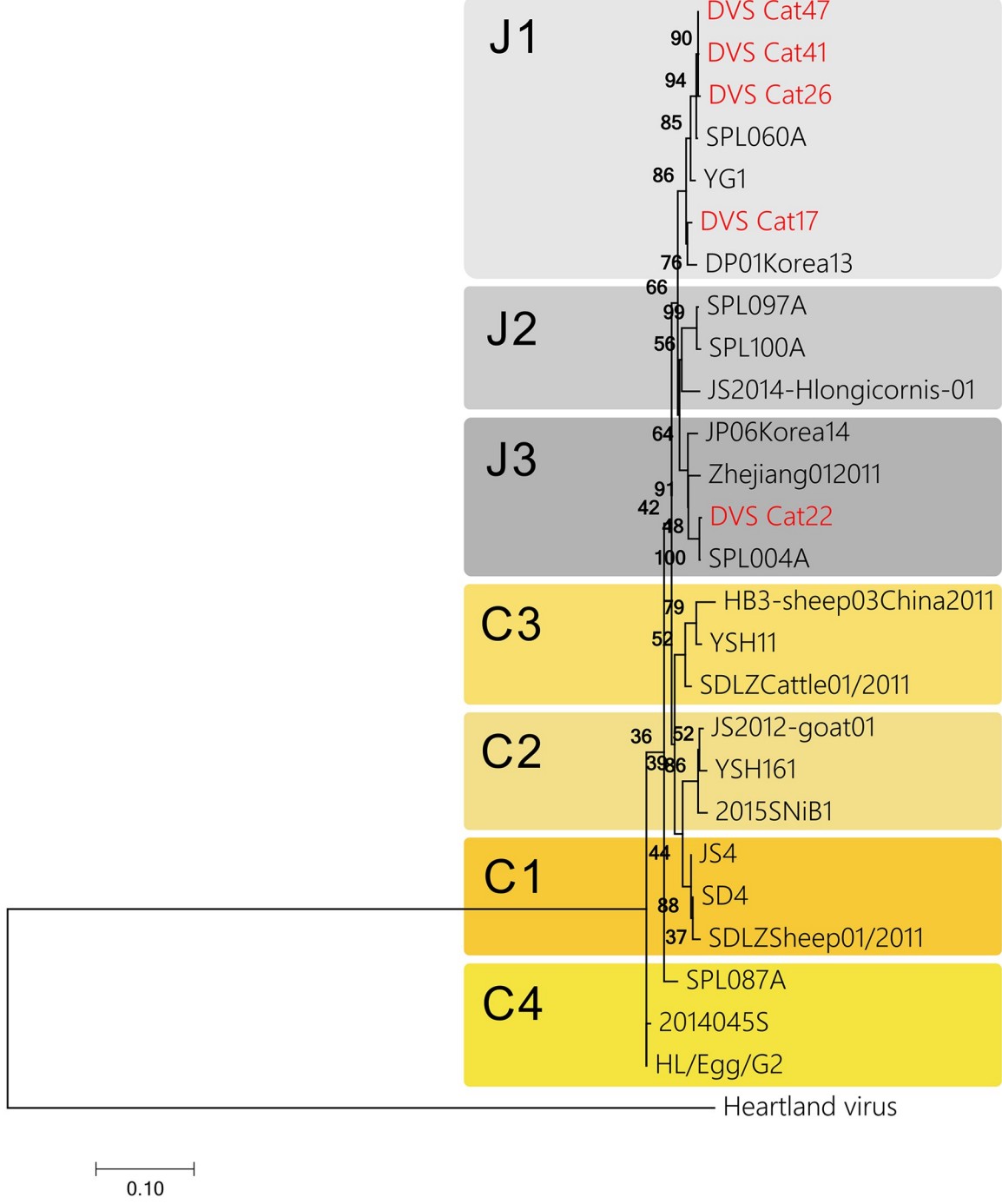

**Fig 4. Phylogenic trees of detected SFTSV genome in this study (red) and reference SFTSV genome (black) for the S segments.**

robust immune response, the samples collected in this study might be too early to assess the neutralizing antibodies in the naturally infected animals.

In conclusion, the RT-PCR approach developed in the present study and the IgM- and IgG-ELISA performed to detect SFTS-specific antibodies were useful for making a laboratory diagnosis of SFTS-suspected cats and dogs.

**Table 4. Distribution of positive cases.**

| | | Area | |
|---|---|---|---|
| | | **SFTS patients reported** | **SFTS patients not reported** |
| Animal cases | Negative cases | | |
| | Cats | 19 | 12 |
| | Dogs | 7 | 12 |
| | Positive cases | | |
| | Cats | 5 | 0 |
| | Dogs | 0 | 0 |

## Supporting information

**S1 Fig. Phylogenic trees of the SFTSV genome for the S (A) and M (B) segments.** Strains that were identified in China, Japan and Korea are indicated by red, blue and red, respectively. (TIF)

## Author Contributions

**Conceptualization:** Eun-sil Park, Masayuki Saijo, Ken Maeda, Shigeru Morikawa.

**Data curation:** Eun-sil Park, Koichi Imaoka.

**Formal analysis:** Eun-sil Park, Osamu Fujita, Masanobu Kimura, Akitoyo Hotta.

**Funding acquisition:** Masayuki Saijo, Ken Maeda, Shigeru Morikawa.

**Investigation:** Eun-sil Park, Osamu Fujita, Masanobu Kimura, Akitoyo Hotta.

**Methodology:** Eun-sil Park, Masayuki Shimojima.

**Project administration:** Koichi Imaoka, Masayuki Saijo, Ken Maeda, Shigeru Morikawa.

**Resources:** Masayuki Shimojima.

**Supervision:** Shigeru Morikawa.

**Validation:** Eun-sil Park, Shigeru Morikawa.

**Writing – original draft:** Eun-sil Park.

**Writing – review & editing:** Koichi Imaoka, Masayuki Shimojima, Masayuki Saijo, Ken Maeda, Shigeru Morikawa.

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
