## [Decision Letter · Decision Letter 0]

28 Oct 2020

PONE-D-20-25871

Development of a reverse transcription polymerase chain reaction for the detection of severe fever with thrombocytopenia syndrome virus from suspected infected animals

PLOS ONE

Dear Dr. Morikawa,

Thank you for submitting your manuscript to PLOS ONE. After careful consideration, we feel that it has merit but does not fully meet PLOS ONE’s publication criteria as it currently stands. Therefore, we invite you to submit a revised version of the manuscript that addresses the points raised during the review process. Please find following review comments by this academic editor (see Additional Editor Comments) and those by an expert reviewer.

We look forward to receiving your revised manuscript.

Kind regards,

Tetsuro Ikegami

Academic Editor

PLOS ONE

Journal Requirements:

4. Please include your tables as part of your main manuscript and remove the individual files. Please note that supplementary tables (should remain/ be uploaded) as separate "supporting information" files

Additional Editor Comments:

[Independent review by academic editor]

The manuscript entitled “Development of a reverse transcription polymerase chain reaction for the detection of severe fever with thrombocytopenia syndrome virus from suspected infected animals” describes the development of conventional RT-PCR for the detection of S-segment (125 or 178bp) or M-segment (185 or 179bp) of SFTSV. The new RT-PCR detected SFTSV RNA from 3 positive controls (SFTSV culture sup: J1, C4, and C3 clades) and 5 of 56 animal samples (dogs and cats), which were IgM and/or IgG positive for SFTSV. Positive clinical samples were all from cats, and the partial S-segment nucleotide sequences of those positive samples were phylogenetically grouped into J1 (Cat47, 41, 26, 17) or J3 (Cat22) clades. This study showed the presence of viremic cats harboring SFTSV in Japan.  

Major points:

The title of this study does not reflect additional SFTS case identification in cats in Japan. The study in this manuscript identified five cats with detectable SFTSV RNA, and specific IgG / IgM. The development of conventional RT-PCR not sufficiently novel for publication by itself. Please consider changing the title, while revising the abstract and conclusion accordingly.Figure 1 and 2 do not provide the rationale of given primer designs, which should be removed or moved to Supplementary materials, if required. The alignment of each primer sequence to representative strains in those clades might be alternatively shown.Line 59: Authors should provide the number of animals, sampling locations, and status of cat and dog cases in this study.Table 3: Information of five cat cases should be individually listed in the Table.Table 4 requires further clarification. It is not clear about “type of assay”, “animal species”, and “type of samples” from the Table information. Corresponding text in line 179 is not apparently relevant to the Table information.In Discussion section, authors should describe current diagnostic options for SFTSV in humans and animals with citations. Although this study used conventional RT-PCR, it is important to provide the rationale to use the classical approach over the real-time PCR.

Minor points:

Figure 3 and 4 are not labelled for size or samples. Authors may combine two images in a single Figure.Line 86-88: Authors should describe the preparation of RNA samples with serial RNA copy numbers: i.e., in vitro transcribed RNA with known copy numbers? If the reference control is DNA, the measurement indicates the cDNA detection limit.Please describe about the content of negative controls in the Materials and Methods and the Figure 3, 4, and 5.Line 133: “RT-PCR was performed to detect in clinical animal specimen” should be deleted, because this section is described about IgM and IgG detection.Authors should clarify if they also tried to isolate SFTSV from those cat samples.Additional analysis of M-segment or other genetic information (e.g., NGS) should better characterize viral isolates. It might be reassortant strains if several strains are co-circulating within the region.Please provide further information of positive cats, in terms of the follow-up isolation from owners or treatment after the diagnosis, in Discussion section.

Reviewers' comments:

Reviewer's Responses to Questions

**Comments to the Author**

1. Is the manuscript technically sound, and do the data support the conclusions?

Reviewer #1: Yes

2. Has the statistical analysis been performed appropriately and rigorously? 

Reviewer #1: N/A

3. Have the authors made all data underlying the findings in their manuscript fully available?

Reviewer #1: Yes

4. Is the manuscript presented in an intelligible fashion and written in standard English?

Reviewer #1: Yes

5. Review Comments to the Author

Reviewer #1: In this article, the authors describe the development and characterization of 2 sets of primers that detect SFTSV in cell supernatants and animal samples. This is a straightforward paper with strong methodology and is fairly well written, with only some minor grammatical changes required.

One change I would like to see is a better description of the impact and contribution of this assay. For example, some of the points that could be added include:

- please include a paragraph describing the clinical presentation of the cats in the study, the timelines of presentation and sampling, and whether they survived or not.

- Table 3 - since there are only five cats it would be nice to display data for each individual cat

- please describe why the authors developed this assay. Does the human rtPCR assay not work in animals? Does the animal rtPCR developed by the authors work in human samples?

- why was the L segment not chosen as a PCR target? It is highly conserved and would be a good choice for detecting as many clades as possible

- please provide background and discussion about how many serotypes exist for SFTSV. How do you know which clade to use for the neutralizing assays? Can antibodies from one clades cross-neutralize other clades? Since you only used one clade, could a serotype mismatch be a possible reason for the lack of neutralization?

Line 16 - remove the word "to"

Line 17 - change "shows" to "have"

Line 21 - change "by" to "from"

Line 23 - remove "were"

Line 23 - change "several" to the specific number of clades the PCR detects

Line 24 - change "by" to "using"

Line 29 - change "clades from" to "clades and from"

Line 25 - change to "where a high number of SFTS patients have been reported, indicating circulating SFTSV in the environment. This diagnostic assay could be a helpful tool in detecting SFTS in animals."

Line 51-52 - I would move this sentence to Line 42. Change to "...are reportedly spreading throughout..."

Line 54 - in your abstract you say the assay worked against several clades, here you say all clades. please pick the correct one and then be consistent.

Line 60 - delete "the"

Lines 86-88 - delete. these are results and do not belong in the methods section.

Line 98 - change to "Sequencing and phylogenetic analysis was performed on the viral S gene segment for all RT-PCR positive samples."

Line 159 - change "nucleotide" to "nucleotide sequence"

Line 163 - delete "according to previous studies"

Line 165 - change "positive on RT-PCR" to "PCR-positive"

Line 172 - the antibodies are still functional, they just are not neutralizing. please modify this.

Line 178 - please include a figure of the four positive bands for the five cats

Line 189 - neutralizing antibodies rarely appear before 7 days after infection, and it usually takes 10-14 days to see a robust response. in this study, your samples were taken too early to really assess whether neutralizing antibodies were induced.

6. PLOS authors have the option to publish the peer review history of their article (what does this mean?). If published, this will include your full peer review and any attached files.

Reviewer #1: No

---

## [Author Response · Author response to Decision Letter 0]

20 Nov 2020

Editor comments

Major points:

 The title of this study does not reflect additional SFTS case identification in cats in Japan. The study in this manuscript identified five cats with detectable SFTSV RNA, and specific IgG / IgM. The development of conventional RT-PCR not sufficiently novel for publication by itself. Please consider changing the title, while revising the abstract and conclusion accordingly.

 1. Figure 1 and 2 do not provide the rationale of given primer designs, which should be removed or moved to Supplementary materials, if required. The alignment of each primer sequence to representative strains in those clades might be alternatively shown.

Answer : Primers in this study were designed using all strains of figure 1 and 2. Since the consensus sequence for the primer design were made with these strains, we showed the phylogenic tree (figure 1) to cover all clades. Figure 2 might be moved to supplementary material. 

 2. Line 59: Authors should provide the number of animals, sampling locations, and status of cat and dog cases in this study.

Answer : The number of animals were added, cats are 37 cases and dogs are 19 cases (line 62). Samples were sent from all around Japan and we described the sites using maps (Figure 2). Also we added the status of dog cases as “Dogs showed anorexia, depression, fever and some gastrointestinal tract symptoms, including diarrhea and vomit (line 65-66). 

3. Table 3: Information of five cat cases should be individually listed in the Table.

Answer : Information of five cat cases were listed individually in the Table 3.

4. Table 4 requires further clarification. It is not clear about “type of assay”, “animal species”, and “type of samples” from the Table information. Corresponding text in line 179 is not apparently relevant to the Table information.

Answer : We changed the Table 4.

 5. In Discussion section, authors should describe current diagnostic options for SFTSV in humans and animals with citations. Although this study used conventional RT-PCR, it is important to provide the rationale to use the classical approach over the real-time PCR.

Answer : We added sentences in discussion section. “In Japan, for the diagnosis of human and animal SFTS cases, conventional one-step RT-PCR to detect the SFTSV genome, ELISA to detect the antibodies against SFTSV and the isolation of SFTSV are performed. However, non-specific reactions were obtained from some SFTS-suspected animal cases, especially dogs, using primers for human SFTS diagnosis. For the simple, rapid and specific detection of SFTSV RNA from SFTS-suspected animal cases, primer sets were newly designed in this study.” (Line 186-191) 

Minor points:

1. Figure 3 and 4 are not labelled for size or samples. Authors may combine two images in a single Figure.

Answer : Figure 3 and 4 were labelled and combined as a single figure.

2. Line 86-88: Authors should describe the preparation of RNA samples with serial RNA copy numbers: i.e., in vitro transcribed RNA with known copy numbers? If the reference control is DNA, the measurement indicates the cDNA detection limit.

Answer : Because the copy numbers of RNA samples have been measured in our previous study, RNA samples with known copy numbers were diluted by 10-fold. We corrected sentences as “RNAs from three strains of SFTSV belonging to different clades - SPL010 (J1 clade, accession No. AB817999), cat#1 (C4 clade, accession No. DRA007207) and HB29 (C3 clade, accession No. NC_018137) - were used as positive controls. The copy numbers of RNA samples used as positive controls have been measured by real time RT-PCR according to our previous study. Then, RNA were diluted by 10-fold with diethylpyrocarbonate (DEPC)-treated distilled water (DW).” (Line 82-87)

3. Please describe about the content of negative controls in the Materials and Methods and the Figure 3, 4, and 5.

Answer : The content of negative control was added. “DEPC-treated DW was used as a negative control.” (Line 87)

4. Line 133: “RT-PCR was performed to detect in clinical animal specimen” should be deleted, because this section is described about IgM and IgG detection.

Answer : This sentence were deleted.

5. Authors should clarify if they also tried to isolate SFTSV from those cat samples.

Answer : We did not try to isolate SFTSV from those positive samples.

6. Additional analysis of M-segment or other genetic information (e.g., NGS) should better characterize viral isolates. It might be reassortant strains if several strains are co-circulating within the region.

Answer : It is of interest to clarify if the genomic reassortment event occurred in animal derived SFTS viruses, however, this is not an purpose of the manuscript so that the following sentence was added in Discussion section,. i.e.,

“In Japan, several phylogenetic clades are circulating in the SFTS endemic region, thus it is of interest to clarify if the genomic reassortment event occurred in animal derived SFTS viruses in future." (Line 198-201)

7. Please provide further information of positive cats, in terms of the follow-up isolation from owners or treatment after the diagnosis, in Discussion section.

Answer : We added the information in Discussion section, i.e., “After cats were diagnosed as SFTS, veterinarians were informed; 1) they should wear personal protective equipment (PPE), such as gloves, mask, face shield, goggle and gown, to treat SFTS-diagnosed cats since SFTSV could be secreted from blood, saliva, urine and feces for about two or three weeks, 2) they should consult a doctor if they have a direct contact with these blood, saliva, urine and feces and then have clinical symptoms, such as, fever, 3) they should inform owners of the animals not to contact without PPE.” (Line 205-210).

Review Comments to the Author

Reviewer #1: In this article, the authors describe the development and characterization of 2 sets of primers that detect SFTSV in cell supernatants and animal samples. This is a straightforward paper with strong methodology and is fairly well written, with only some minor grammatical changes required.

1. please include a paragraph describing the clinical presentation of the cats in the study, the timelines of presentation and sampling, and whether they survived or not.

Answer : We added sentences as “Five SFTS cat cases showed clinical symptoms including depression, loss of appetite and jaundice. They had body weight loss, fever, leukocytopenia (2180~7540/μL), thrombocytopenia (15,000~120,000/μL) and high total bilirubin level. All of five cats were indoor and outdoor and four of five cats had tick-bite history. Samples were collected within one week after disease onset. Two cats were dead and three cats recovered (Table 3).” (Line 165-169).

2. Table 3 - since there are only five cats it would be nice to display data for each individual cat

Answer : We added the information of five cats in table 3. 

3. please describe why the authors developed this assay. Does the human rtPCR assay not work in animals? Does the animal rtPCR developed by the authors work in human samples?

Answer : Some researchers including us tried the human rtPCR assay to detect the animal SFTS cases. However, there were lots of non-specific results, especially in dogs. We performed the sequence analysis those cases, and found that some genomes of dogs could react in the human rtPCR assay. We described that in Discussion section (line 186-191).

4. why was the L segment not chosen as a PCR target? It is highly conserved and would be a good choice for detecting as many clades as possible

Answer : Because we thought that two segments are enough to diagnose animal SFTS cases, we chose two segments, S segment and M segment.

5. please provide background and discussion about how many serotypes exist for SFTSV. How do you know which clade to use for the neutralizing assays? Can antibodies from one clades cross-neutralize other clades? Since you only used one clade, could a serotype mismatch be a possible reason for the lack of neutralization?

Answer : We added about how many serotypes exist for SFTSV (line43-44). We tried the neutralization antibody assay using neutralizing antibody positive sera and SFTSV strain belonging to other clades. Although the rate of neutralization in one dilution point was different a little, the titer of neutralization antibody was not different. So, we performed the final neutralization antibody assay using one clade and described the results. Anyway, the titer of neutralizing antibody of five cats were under the limit of detection.

6. Line 16 - remove the word "to"

Answer : We removed the word “to”.

7. Line 17 - change "shows" to "have"

Answer : We changed “show” to “have”.

8. Line 21 - change "by" to "from"

Answer : We changed “by” to “from”.

9 . Line 23 - remove "were"

Answer : We removed “were”.

10. Line 23 - change "several" to the specific number of clades the PCR detects

Answer : We changed “several” to “six”.

11. Line 24 - change "by" to "using"

Answer : We changed “by” to “using”.

12. Line 29 - change "clades from" to "clades and from"

Answer : We changed “clades from” to “clades and from”.

13. Line 35 - change to "where a high number of SFTS patients have been reported, indicating circulating SFTSV in the environment. This diagnostic assay could be a helpful tool in detecting SFTS in animals."

Answer : We changed the sentence.

14. Line 51-52 - I would move this sentence to Line 42. Change to "...are reportedly spreading throughout..."

Answer : We moved the sentence to Line 43 and changed. 

15. Line 54 - in your abstract you say the assay worked against several clades, here you say all clades. please pick the correct one and then be consistent.

Answer : We changed to “several clades” to be consistent.

16. Line 60 - delete "the"

Answer : We deleted “the”.

17. Lines 86-88 - delete. these are results and do not belong in the methods section.

Answer : We deleted these.

18. Line 98 - change to "Sequencing and phylogenetic analysis was performed on the viral S gene segment for all RT-PCR positive samples."

Answer : We changed as your advice (line 104-105).

19. Line 159 - change "nucleotide" to "nucleotide sequence"

Answer : We changed to “nucleotide sequence” (line 170).

 20. Line 163 - delete "according to previous studies"

Answer : We deleted the phrase.

21. Line 165 - change "positive on RT-PCR" to "PCR-positive"

Answer : We changed to “PCR-positive” (line 176).

22. Line 172 - the antibodies are still functional, they just are not neutralizing. please modify this.

Answer : We changed to “could not neutralize SFTSV” (line 183).

23. Line 178 - please include a figure of the four positive bands for the five cats

Answer : We included all five positive bands in the Figure 5.

24. Line 189 - neutralizing antibodies rarely appear before 7 days after infection, and it usually takes 10-14 days to see a robust response. in this study, your samples were taken too early to really assess whether neutralizing antibodies were induced.

Answer : We changed the sentence like this.

“It takes approximately two weeks to see a robust immune response, the samples collected in this study might be too early to assess the neutralizing antibodies in the naturally infected animals.” (Line 213-215)

---

## [Editor Report · Decision Letter 1]

2 Dec 2020

PONE-D-20-25871R1

Diagnostic system for the detection of severe fever with thrombocytopenia syndrome virus from suspected infected animals

PLOS ONE

Dear Dr. Morikawa,

Thank you for submitting your manuscript to PLOS ONE. After careful consideration, we feel that it has merit but does not fully meet PLOS ONE’s publication criteria as it currently stands. Therefore, we invite you to submit a revised version of the manuscript that addresses the points raised during the review process.

The revised manuscript entitled “Diagnostic system for the detection of severe fever with thrombocytopenia syndrome virus from suspected infected animals” significantly improve the presentation of contents. Please consider a few additional minor changes.

Minor points:

Line 15: “those of SFTS patients, whereas SFTS-contracted cats….”Line 22: “specifically detected six clades of SFTSV” in the Abstract was not shown in the Result section. Please consider the revision.Line 35: “indicating the cat-to-human transmission of SFTSV”?Line 46: a variety of animals, while viral RNA and antibodies…”Line 64: “vomiting”The rationale of new RT-PCR appeared in Discussion (lines 182 – 187). A similar brief rationale should be given in Introduction and Abstract sections.Title: “Diagnostic system for the detection of severe fever with thrombocytopenia syndrome virus RNA from suspected infected animals” or “Diagnostic system for the RNA detection of severe fever with thrombocytopenia syndrome virus from suspected infected animals”

We look forward to receiving your revised manuscript.

Kind regards,

Tetsuro Ikegami

Academic Editor

PLOS ONE

---

## [Author Response · Author response to Decision Letter 1]

4 Dec 2020

1. Line 15: “those of SFTS patients, whereas SFTS-contracted cats….”

Response : We corrected the sentence according to the comment (line 15-16). 

2. Line 22: “specifically detected six clades of SFTSV” in the Abstract was not shown in the Result section. Please consider the revision.

Response : “six clades” were changed to “four clades” since two clades were not included in the clinical samples in the present study. However, the primers designed in the completely conserved region among eight clades, thus the PCR might detect all the eight clades. In this regard, we added the contents in the Result and Discussion section (line 173-174, 192-194). 

3. Line 35: “indicating the cat-to-human transmission of SFTSV”?

Response : The sentence was corrected according to the comment (line 36).

4. Line 46: a variety of animals, while viral RNA and antibodies…”

Response : The sentence was corrected according to the comment (line 46).

5. Line 64: “vomiting”

Response : The sentence was corrected according to the comment (line 67).

6. The rationale of new RT-PCR appeared in Discussion (lines 182 – 187). A similar brief rationale should be given in Introduction and Abstract sections.

Response : The brief rationale were added in the Abstract and Introduction sections (line 18-19, 53-55). 

7. Title: “Diagnostic system for the detection of severe fever with thrombocytopenia syndrome virus RNA from suspected infected animals” or “Diagnostic system for the RNA detection of severe fever with thrombocytopenia syndrome virus from suspected infected animals”

Response : Title was changed to “Diagnostic system for the detection of severe fever with thrombocytopenia syndrome virus RNA from suspected infected animals”.

---

## [Editor Report · Decision Letter 2]

7 Dec 2020

Diagnostic system for the detection of severe fever with thrombocytopenia syndrome virus RNA from suspected infected animals

PONE-D-20-25871R2

Dear Dr. Morikawa,

We’re pleased to inform you that your manuscript has been judged scientifically suitable for publication and will be formally accepted for publication once it meets all outstanding technical requirements.

Kind regards,

Tetsuro Ikegami

Academic Editor

PLOS ONE

---

## [Editor Report · Acceptance letter]

15 Dec 2020

PONE-D-20-25871R2 

Diagnostic system for the detection of severe fever with thrombocytopenia syndrome virus RNA from suspected infected animals 

Dear Dr. Morikawa:

I'm pleased to inform you that your manuscript has been deemed suitable for publication in PLOS ONE. Congratulations! Your manuscript is now with our production department. 

Kind regards, 

on behalf of

Dr. Tetsuro Ikegami 

Academic Editor

PLOS ONE